# Back to GroEL-Assisted Protein Folding: GroES Binding-Induced Displacement of Denatured Proteins from GroEL to Bulk Solution

**DOI:** 10.3390/biom10010162

**Published:** 2020-01-20

**Authors:** Victor Marchenkov, Andrey Gorokhovatsky, Natalia Marchenko, Tanya Ivashina, Gennady Semisotnov

**Affiliations:** 1Institute of Protein Research, Russian Academy of Sciences, Institutskaya Street 4, 142290 Pushchino, Russia; march@phys.protres.ru (V.M.); lita@phys.protres.ru (N.M.); 2Shemyakin-Ovchinnikov Institute of Bioorganic Chemistry, Russian Academy of Sciences, Ulitsa Miklukho-Maklaya 16/10, GSP-7, 117997 Moscow, Russia; andrey.gorokhovatsky@yandex.ru; 3Skryabin Institute of Biochemistry and Physiology of Microorganisms, Russian Academy of Sciences, Prospect Nauki 5, 142290 Pushchino, Russia; ivashina@ibpm.pushchino.ru

**Keywords:** GroEL-assisted protein folding, chaperonins, protein-protein interaction, bioluminescence, fluorescence

## Abstract

The main events in chaperone-assisted protein folding are the binding and ligand-induced release of substrate proteins. Here, we studied the location of denatured proteins previously bound to the GroEL chaperonin resulting from the action of the GroES co-chaperonin in the presence of Mg-ATP. Fluorescein-labeled denatured proteins (α-lactalbumin, lysozyme, serum albumin, and pepsin in the presence of thiol reagents at neutral pH, as well as an early refolding intermediate of malate dehydrogenase) were used to reveal the effect of GroES on their interaction with GroEL. Native electrophoresis has demonstrated that these proteins tend to be released from the GroEL-GroES complex. With the use of biotin- and fluorescein-labeled denatured proteins and streptavidin fused with luciferase aequorin (the so-called streptavidin trap), the presence of denatured proteins in bulk solution after GroES and Mg-ATP addition has been confirmed. The time of GroES-induced dissociation of a denatured protein from the GroEL surface was estimated using the stopped-flow technique and found to be much shorter than the proposed time of the GroEL ATPase cycle.

## 1. Introduction

Studies of protein refolding reactions mainly confirm Anfinsen’s hypothesis that the necessary and sufficient information on the spatial structure of proteins is encoded in their amino acid sequences [1,2,3]. Nevertheless, a number of cellular proteins have been revealed to be required for the formation of the protein native conformation [4,5]. Many of these proteins, named molecular chaperones [4], are members of a large group of heat shock proteins (hsps), whose biosynthesis in the cell is enhanced by various cellular stresses [6]. Molecular chaperones facilitate the folding of various proteins both in vivo and in vitro by interacting transiently and non-covalently with non-native early or late folding intermediates and not with the native (rigidly packed) protein. This interaction prevents aggregation of the intermediates or, perhaps, repairs their misfolded conformations. The rigidity of a protein spatial structure can be lost also due to various cellular stresses, mutations, and protease-induced cleavage, thus making the protein prone to aggregation fatal for its recovery or degradation. Accumulation of protein aggregates in various tissues leads to amyloidosis-based diseases [7]. Therefore, the study of structural and functional properties of chaperones is of primary importance. Most studied, both structurally and functionally, are the GroEL (hsp60) and GroES (hsp10) molecular chaperones of *Escherichia coli* often called chaperonins [8]. The electron microscopy [9,10] and crystallography [11,12] data show that 14 identical subunits (57 kDa each) of the GroEL chaperonin are arranged into two stacked heptameric toroids forming a cylinder with a central inner cavity of 45 Å in diameter.

Each GroEL subunit consists of three distinct domains: the apical domain responsible for binding the substrate polypeptides and the GroES co-chaperonin, the equatorial domain providing most of the intersubunit interactions and binding the ligands (Mg ions and adenyl nucleotides – ATP and ADP), and the intermediate (“hinge”) domain providing allosteric conformational changes of GroEL that result from the binding of ligands [13] and GroES [14]. The GroES co-chaperonin consists of 7 identical subunits (~10 kDa each) arranged into an annular dome-shaped quaternary structure [15]. In the presence of Mg-ATP or Mg-ADP, GroES interacts with GroEL apical domains to form either an asymmetric (with one GroES) [14] or symmetric (with two GroESs) [10] complex. In this case, the inner cavity of the complex substantially increases due to a change in the position of the mobile apical and intermediate domains of GroEL, as well as an additional inner cavity of GroES [14]. The presence of an extensive inner cavity and the high mobility of the apical domains interacting with substrate proteins are the basis for the proposal of two models of the GroEL-assisted folding of other proteins [16,17,18,19,20,21].

One of these, the “active” model [18,20,21], implies the direct involvement of GroEL (and other chaperonins) in the “folding by forced unfolding”, i.e., participation in the unfolding of misfolded protein conformations, thereby promoting their correct refolding.

The other (“passive”) model [16,17,19] suggests that substrate proteins fold in the central cavity of the GroEL•GroES chaperonin complex (“Anfinsen cage”) where they are free from undesirable intermolecular contacts and can assume their native (rigidly packed) conformation. The location of a substrate protein in the cavity and its release are regulated by the GroEL ATPase activity [17]. The time of each binding/release cycle has been estimated as 10–15 s (see for references [17,22]).

Despite experimental evidence for validity of both models, they seem to be at least not universal. Firstly, the possible unfoldase activity of GroEL [23] is not clearly connected with the acceleration of the substrate protein folding [24]. Secondly, the inner cavity of GroEL and other chaperonins is not a necessary prerequisite for the chaperone activity because many chaperones without such a cavity (for example, Hsp70 and Hsp90) [5,6] and even solely GroEL apical domains [25,26,27] display an activity similar to that of chaperonins.

The analysis of the literature data on GroEL interaction with substrate proteins [28] and on the GroEL effect on the refolding kinetics of a number of globular proteins [29,30,31,32] allow us to propose another version of the “passive” model that implies no intracavity protein folding [28]. To inhibit non-productive aggregation of a substrate protein, it is sufficient to substantially reduce its concentration in bulk solution via its interaction with the GroEL apical domains (a similar approach was proposed by Buchner et al. [33]). The strength of this interaction is determined by the balance of hydrophobic and electrostatic forces, which is regulated by ligands and the GroES co-chaperonin. If the affinity of the substrate protein for GroEL is not high, the latter can assist its folding without recruiting the ligands and GroES (as described in [34,35]). However, the presence of the oligomeric ring-shaped structure, and hence, many binding sites (apical domains) in most cases leads to high stability (a long lifetime) of GroEL complexes with denatured polypeptides [36,37]. The substrate protein must dissociate from the GroEL surface to bulk solution to be able to fold [28]. Mg-ATP, Mg-ADP, and GroES reduce the tightness of GroEL binding to substrate proteins, thereby increasing the probability of their release and folding [30,31].

Here we demonstrate that some relatively small (m.w. ≤ 30 kDa) denatured proteins, such as pepsin, α-lactalbumin, serum albumin, and lysozyme with broken intramolecular disulfide bonds or an early kinetic folding intermediate of malate dehydrogenase, dissociate from the GroEL surface under the influence of Mg-ATP and GroES. To confirm the fact of their dissociation to bulk solution, we used the so-called streptavidin trap (streptavidin fused with aequorin luciferase), fluorescein- and biotin-labeled α-lactalbumin, and malate dehydrogenase (MDH). The binding of these substrate proteins to the streptavidin trap in bulk solution was established by bioluminescence resonance energy transfer (BRET). It turned out that the GroES-induced transfer of the denatured proteins from GroEL to bulk solution occurs in a much shorter time than that proposed for the GroEL ATPase cycle.

## 2. Materials and Methods

### 2.1. Solutions

All solutions were prepared using double distilled water and chemical reagents purchased from commercial firms (Sigma-Aldrich (St. Louis, MO, USA), REANAL Pharmaceuticals and Fine Chemicals Ltd. (Budapest, Hungary), SERVA Electrophoresis GmbH (Heidelberg, Germany), Reachim Ltd., (Moscow, Russia), Bayer AG (Leverkusen, Germany), Boehringer Mannheim GmbH (Mannheim, Germany), Molecular Probes (Eugene, OR, USA), Invitrogen (Waltham, MA, USA)) without additional purification.

### 2.2. Proteins

The recombinant GroEL and GroES chaperonins were purified after expression in *E. coli* cells (strain HB101) the multicopy plasmid pGroE4 (groE operon of *E. coli* cloned in the EcoR1 site of the pACYC184 vector) according to the published protocols [38,39]. Protein concentrations were measured spectrophotometrically using the known extinction coefficients (A^0.1%^_1 cm_) 0.19 for GroEL and 0.14 for GroES [38,40].

The plasmid (pETj/SAV-L-Aeq) encoding streptavidin fused with aequorin was prepared and expressed in *E. coli* cells BL21 (DE3) as described previously [41] with detailed methods of purification of the fusion protein and measuring of its bioluminescence spectrum.

Substrate proteins (bovine α-lactalbumin, white egg lysozyme, bovine pepsin, bovine serum albumin, and pig mitochondrial malate dehydrogenase) purchased from Sigma or Serva were additionally purified by ion-exchange and size-exclusion chromatography. Fluorescein-labeled proteins (5–10 mg/mL) were prepared by incubation in 0.2 M NaHCO_3_ buffer, pH 8.4, with a 10-fold molar excess of 5(6)-FAM, SE (5-(and-6)-Carboxyfluorescein, Succinimidyl Ester) in dimethyl sulfoxide (DMSO) for 1 h at 23 °C with intensive stirring. The reaction was stopped by the addition of Tris-HCl, pH 7.5, of a concentration up to 10 mM. The additional purification of fluorescein-labeled proteins from free labels was performed using a PD10 (Sephadex G25) column and the same buffer. Further purification and segregation of molecules with different numbers of labels were performed by DEAE-ToyoPearl or CM-ToyoPearl chromatography. The concentration of fluorescein-labeled proteins was determined using an amino acid analyzer LC 7000 (Biotronic, Germany). The number of fluorescein labels was determined using the known fluorescein extinction coefficient A^1M^_490 nm_ = 87,000 o.u. The fractions of fluorescein-labeled proteins containing 1 label per 1 protein molecule were used to monitor their binding to and dissociation from GroEL. Biotinylation of the purified fluorescein-labeled proteins was performed by the same techniques using biotin succinimide ester. 

### 2.3. Methods

Denaturing and non-denaturing electrophoresis were performed as described previously [39]. The aggregation of fluorescein-labeled lysozyme was initiated by the addition of 10 mM DTT to 10 µM protein solution in 20 mM Tris-HCl, pH 7.5, 10 mM MgCl_2_ both in the absence and in the presence of equimolar GroEL, GroES, and 10 mM ADP. After certain incubation time, the mixtures were centrifugated for 10 min at 12,000× *g* and fluorescence spectra of supernatants were measured. The protein solution without DTT was used as a standard for fluorescence intensity. Fluorescence spectra and anisotropy, as well as bioluminescence, were measured using a Cary Eclipse spectrofluorimeter (Varian Medical Systems, Palo Alto, CA, USA). Static and manual-mixing kinetic measurements were performed in a standard 1 × 1 × 4 cm quartz cell using a magnetic stirrer. A stopped-flow device combined with a Chirascan spectropolarimeter (Applied Photophysics Ltd, London, UK) was used for the registration of fast kinetics. The approximation of the kinetic curves was made using the SigmaPlot computer program (Systat Software Inc., Chicago, IL, USA).

The absorption spectra were recorded using a Cary 100 Bio spectrophotometer (Varian Medical Systems, Palo Alto, CA, USA).

## 3. Results

### 3.1. GroES Essentially Decreases the GroEL Affinity for Denatured Proteins

The experiments on the release of substrate proteins from the GroEL surface were mainly performed using various protein refolding systems (see, for example, [29,30,31,32,33]). However, in this case, the events of the formation of the native structure of target proteins and their release from GroEL are indiscernible because native proteins usually do not bind to GroEL. Nevertheless, when the formation of the target protein native structure is inhibited by experimental conditions (disruption of intramolecular disulfide bonds, non-native pH or temperature) or by mutations or truncation of the protein chain, it becomes possible to distinguish between the protein binding to and its dissociation from GroEL.

Figure 1 represents the non-denaturing electrophoresis of the GroEL complexes with fluorescein-labeled denatured proteins in the presence of DTT and Mg-ADP before (slot 1) and after (slot 2) 10 min incubation with a two-fold molar excess of GroES. 

It is seen that even in the presence of Mg-ADP these denatured proteins are tightly bound to GroEL (the presence of fluorescein fluorescence in the GroEL bands). At the same time, after 10 min incubation of these complexes with Mg-ADP and GroES, fluorescein fluorescence is practically absent from the GroEL band. Moreover, the same is true for the band of the GroEL•GroES complex whose motion is slightly slower than that of GroEL alone (cf. slots 1 and 2). Hence, we can assume that denatured proteins are displaced from GroEL by GroES. Note that free denatured fluorescein-labeled proteins have a much higher electrophoretic mobility than GroEL_14_ and GroES_7,_ being positioned at the gel end under these conditions (not shown).

To analyze the dissociation of the denatured proteins from the GroEL surface to bulk solution, we, first of all, established that the thiol-induced cleavage of intermolecular disulfide bonds in lysozyme leads to its aggregation depending on concentration, temperature and time. Figure 2 represents the data on DTT-induced aggregation of lysozyme in the absence and presence of GroEL and GroES. 

As seen, under the chosen conditions, almost entire reduced lysozyme aggregates with a half-time of about 10 min. In contrast, in the presence of equimolar GroEL, the aggregation rate is much lower. Hence, the aggregation of non-native lysozyme is impeded by its interaction with GroEL. However, the presence of ADP, especially together with GroES, increases the rate of reduced lysozyme aggregation.

Next, as shown in Figure 3, we made experiments using streptavidin (heterotetramer) capable of tight biotin-binding [41]. 

α-Lactalbumin and pig mitochondrial malate dehydrogenase (MDH) (GroEL stringent substrate protein) were labeled with biotin and fluorescein, while streptavidin subunits were fused with aequorin (luciferase from *Aequoria victoria*) [41]. Biotin provides tight binding of substrate proteins to streptavidin present in bulk solution, and the fluorescein label is a good acceptor of aequorin bioluminescence. The substrate protein dissociation from the GroEL surface to bulk solution was detected by measuring BRET between aequorin fused with streptavidin and fluorescein dye attached to the biotin-labeled substrate protein. The experiment with the streptavidin trap included the following steps. First, α-lactalbumin in the presence of 5 mM DTT or MDH unfolded in 8 M urea was diluted with buffer containing equimolar native GroEL. An early refolding intermediate of MDH or reduced α-lactalbumin was tightly bound to GroEL and did not interact with streptavidin added later (Figure 3c). Second, 10 s later, 10 mM Mg-ATP and a two-fold molar excess of GroES were added to the mixture. Third, after 5 s incubation, equimolar streptavidin-aequorin was added to the mixture. In this case, aequorin was preliminarily charged with coelenterazine [41]. Fourth, after 5 s incubation, a 10-fold excess of free biotin was added to saturate all free biotin binding sites on streptavidin. After this, Ca-induced bioluminescence of aequorin was measured and the fluorescein component of bioluminescence spectra was analyzed as BRET efficiency (Figure 3d).

The BRET efficiencies evaluated from bioluminescence spectra as the ratio between bioluminescence intensities at 520 nm and 470 nm (I_520_/I_470_) are shown in Figure 4. 

The ratio I_520nm_/I_470nm_ for pure aequorin bioluminescence spectra is 0.55 ± 0.02, which is close (within experimental error) to cases when the interaction of BFL or BFM with streptavidin-aequorin is inhibited (Figure 4 (a), (c), (e), (g)). The larger values (Figure 4 (b), (d), (f), (h)) imply some BRET-induced contribution of fluorescein fluorescence (Figure 3b,d). This contribution mainly depends on the donors/acceptors ratio, the distance between them, and their mutual orientation [41].

The maximal BRET efficiency is exerted by streptavidin-fused aequorin in complex with biotinylated fluorescein-labeled denatured proteins in the absence of GroEL and its ligands. At the same time, when the fusion protein is preliminarily incubated with excessive free biotin, its interaction with denatured proteins is practically inhibited. When GroEL is present in the solution of denatured proteins before the addition of the streptavidin trap, the BRET efficiency is much lower and close to that in the presence of the biotin-saturated streptavidin trap. This is indicative of a strongly restricted accessibility of the streptavidin trap for the GroEL-bound substrate proteins. Thus, the streptavidin trap binds mainly free GroEL-unbound substrate proteins. According to the schemes in Figure 3 and Figure 5, the addition of GroES and Mg-ATP to the complex of GroEL with substrate proteins results in a higher BRET efficiency, which indicates that the substrate proteins dissociate from the GroEL surface to bulk solution, where they bind to the streptavidin trap (Figure 5). 

### 3.2. GroES-Assisted Dissociation of Denatured Proteins from the GroEL Surface Takes Much Less Time than the ATPase Cycle

The above results show that the GroES lid cannot retain a denatured protein in the complex with GroEL. The characteristic binding/release times of a substrate protein under the used conditions were derived from fluorescein fluorescence intensity changes (Figure 6).

Figure 6 represents manual-mixing changes in fluorescence intensity and anisotropy of fluorescein-labeled α-lactalbumin in the presence of 10 mM DTT with successive addition of GroEL and Mg-ATP and GroES. As seen, the addition of equimolar GroEL results in a considerable increase of fluorescence intensity and anisotropy, thereby evidencing for the formation of a complex of the large GroEL molecule with much smaller molecules of denatured α-lactalbumin. But the subsequent addition of 5 mM ATP and a 5-fold molar excess of GroES brings these values back practically to the initial level. Because fluorescein-labeled reduced α-lactalbumin has the typical times of its association with and GroES-induced dissociation from GroEL within a range of 2.5 min in manual-mixing experiments (Figure 6), the stopped-flow technique was used to measure these times.

Figure 7 represents the stopped-flow kinetics of GroEL association/dissociation of fluorescein-labeled reduced α-lactalbumin. As follows from experiments with the streptavidin trap (Figure 3 and Figure 4), under the used conditions, denatured α-lactalbumin binds to GroEL with a half-time of 1.3 s, while its GroES-induced dissociation from GroEL occurs with a half-time of 0.1 s.

## 4. Discussion

Although some evidence that GroEL ligands (ADP, ATP) and the GroES co-chaperonin decrease the chaperonin affinity for non-native proteins have been reported (see, for example [29,30,31]), strong evidence for the displacement of the polypeptides to bulk solution is absent. The results of the current study (Figure 1, Figure 2, Figure 3 and Figure 4) confirm that even relatively small thiol-denatured proteins are displaced from the GroEL surface to bulk solution when the GroES co-chaperonin is bound. After the dissociation, these proteins tend to aggregate (Figure 2) or interact with an external “trap” (Figure 3 and Figure 4).

Since reduced proteins cannot adopt the native structure, their transfer from the GroEL surface to bulk solution is a puzzle. It is generally recognized that the equilibrium must be strongly shifted towards the complex of denatured proteins with GroEL, including the nucleotides and GroES [16,17,19,22]. Note that the ability of a GroES-affected non-native protein to aggregate after dissociation from the GroEL surface to bulk solution was previously shown for 17- and 27-kDa fragments of the elongation factor G [42].

Another surprising result is the time of denatured protein dissociation from the GroEL surface caused by Mg-ATP and GroES (Figure 7b) The half-time of this reaction evaluated from kinetics (Figure 7b), is equal to 0.1 s which is much shorter than the proposed time of the GroEL ATPase cycle (10–15 s [17,22]).

## 5. Conclusions

The studies of GroEL and GroES structural and functional properties have been being in progress for nearly thirty years (see for references [8,17,22,28]), but the main question as to how these chaperonins assist the folding of a broad variety of proteins remains unanswered, although many aspects of this problem have been solved. Firstly, the electron microscopy and crystallographic studies revealed the spatial structures of GroEL [10,11], GroES [15] and their complexes [10,14]. Secondly, it is well established that in the GroEL inner cavity, the binding of substrate polypeptides to apical domains is mediated by multisite interactions (see for references [22]). The affinity of substrate polypeptides for GroEL depends on their hydrophobicity and the electrostatic charge (see for references [17,22,28]). Thirdly, the GroEL ligands (Mg-ATP and Mg-ADP) and the GroES co-chaperonin induce large-scale conformational changes of GroEL that are accompanied by a decrease in its hydrophobicity (see for references [28]). However, it remains unclear where exactly (on the GroEL surface, or in the GroEL/GroES inner cavity, or bulk solution) the folding of substrate proteins occurs, and the role of ATPase in this process remains obscure. The results of our study evidence for the Mg-ATP- and GroES-induced displacement of denatured proteins from GroEL to bulk solution. Surprisingly, the dissociation time appears much shorter than that proposed earlier (see for references [17,22,28]), which offers another aspect of the mechanism of chaperonin-assisted protein folding to be investigated.

## Figures and Tables

**Figure 1 biomolecules-10-00162-f001:**
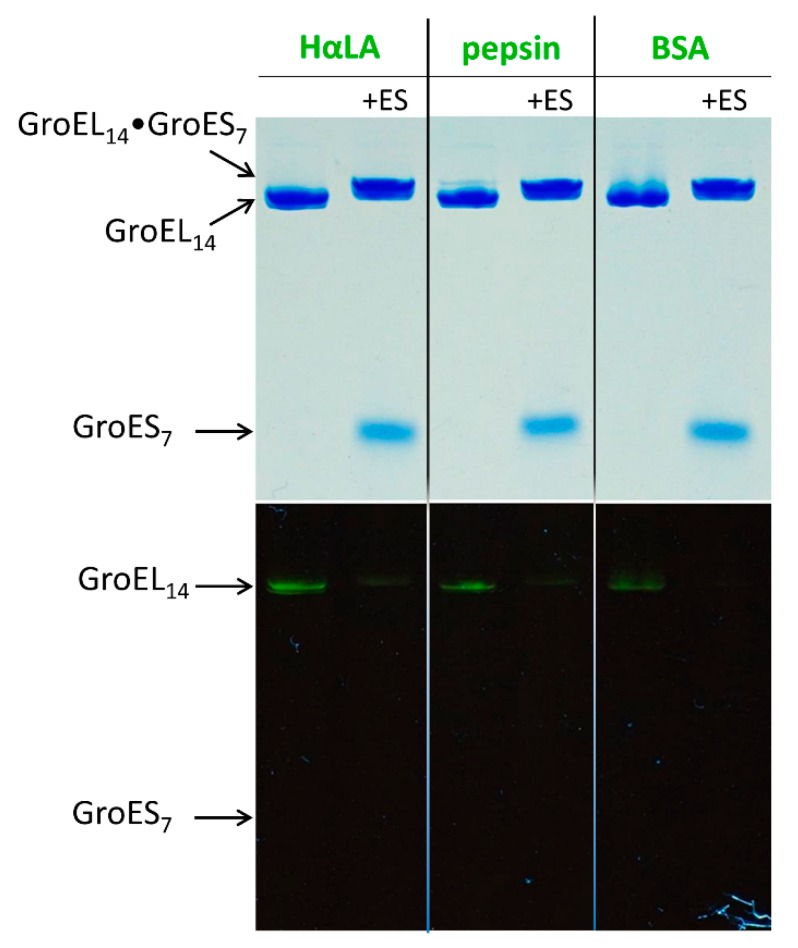
Non-denaturing electrophoresis of the GroEL complex with equimolar fluorescein (FAM)-labeled denatured proteins in the presence of Mg-ADP. In the absence (slot 1) or presence (slot 2) of GroES (two-fold molar excess). Upper part, Coomassie staining; lower part, fluorescence.

**Figure 2 biomolecules-10-00162-f002:**
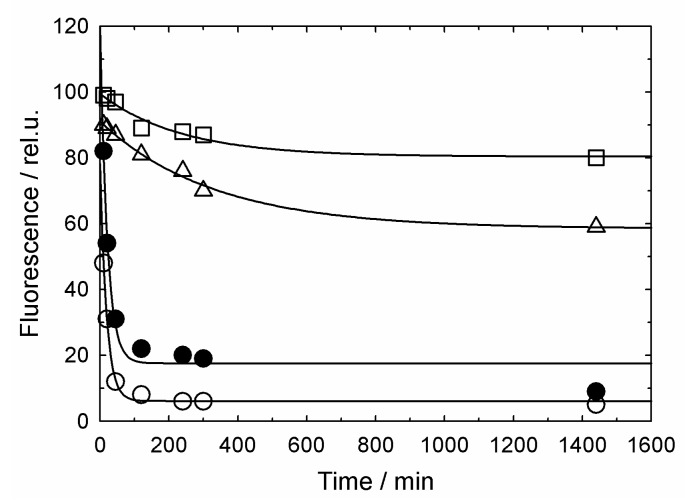
Time-dependent presence of denatured fluorescein-labeled lysozyme in the supernatant (after centrifugation) during the protein incubation in the presence of 5 mM DTT without (o) and with equimolar GroEL (□) in the presence of 10 mM Mg-ADP (∆) and a two-fold molar excess of GroES (●). Solid lines represent an exponential approximation.

**Figure 3 biomolecules-10-00162-f003:**
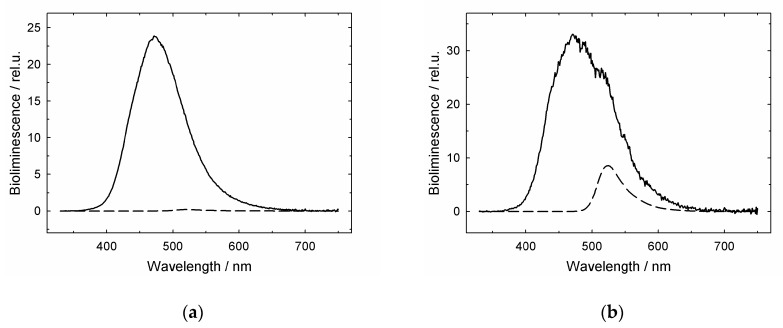
Bioluminescence spectra of streptavidin-aequorin fusion (SAVAq) in the presence of biotin- and fluorescein-labeled α-lactalbumin (BFL), GroEL, ATP + GroES, and free biotin added in the following order: (**a**), biotin + SAVAq→10 s incubation→+BFL; (**b**), SAVAq + BFL→10 s incubation→+free biotin→10 s incubation→+GroEL; (**c**), GroEL + BFL→10 s incubation→+SAVAq→5 s incubation→+free biotin; (**d**), GroEL + BFL→10 s incubation→+10 mM ATP, GroES→5 s incubation→+SAVAq→5 s incubation→+free biotin. Dashed lines denote fluorescein (acceptor) components of aequorin bioluminescence spectra.

**Figure 4 biomolecules-10-00162-f004:**
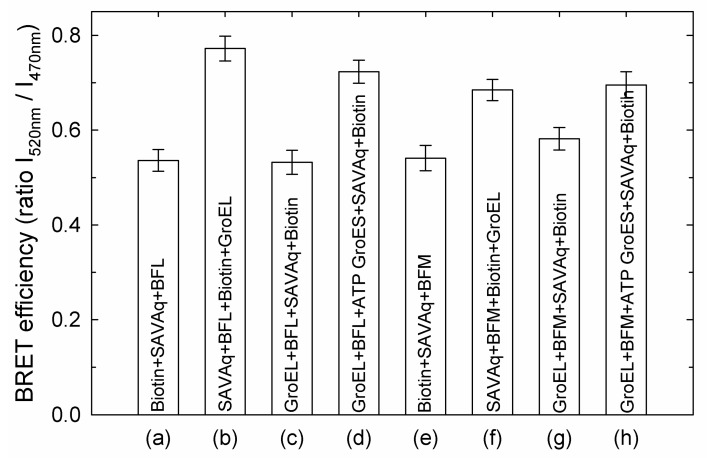
The intensity ratio I_520 nm_/I_470 nm_ for streptavidin-aequorin bioluminescence spectra recorded after manipulations described in Legend to Figure 3. The fluorescein- and biotin-labeled α-lactalbumin (BFL) is lettered as (a), (b), (c), (d), while malate dehydrogenase (BFM) as (e), (f), (g), (h). The arrows (I) show mean-square errors derived from averaging of 10 spectra.

**Figure 5 biomolecules-10-00162-f005:**
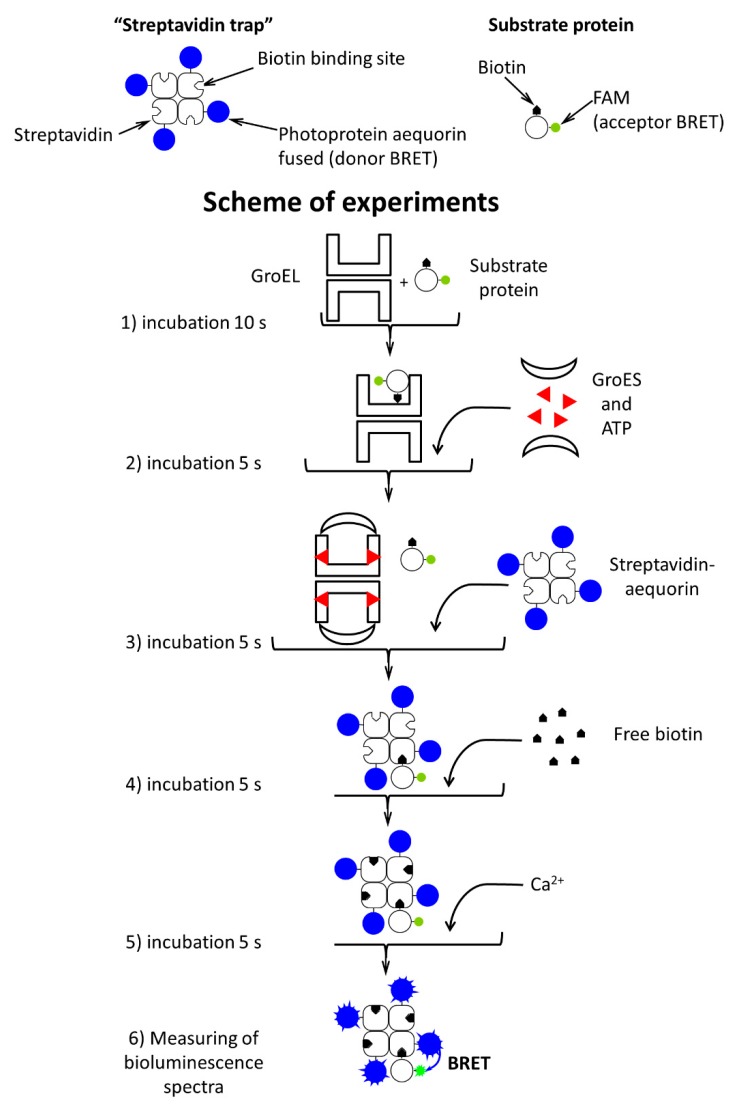
Schematic presentation of experiments using the streptavidin trap.

**Figure 6 biomolecules-10-00162-f006:**
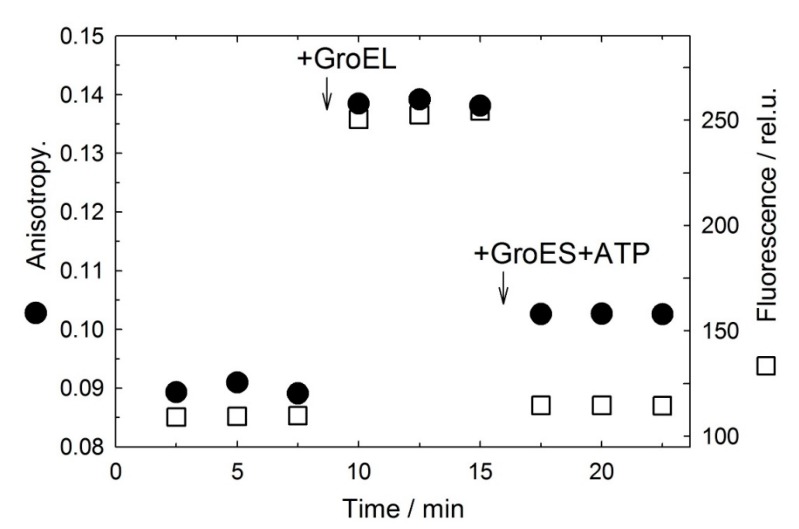
Kinetics of denatured FAM-α-lactalbumin fluorescence intensity at 520 nm (□) and anisotropy (●) changes after successive addition of GroEL and GroES+ATP (shown by arrows).

**Figure 7 biomolecules-10-00162-f007:**
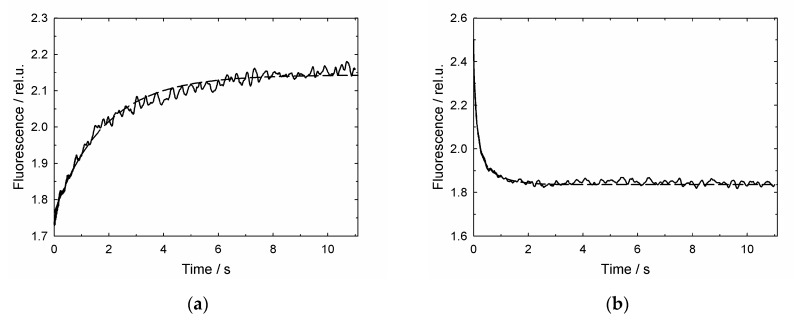
The stopped-flow kinetics of denatured FAM-α-lactalbumin interaction with GroEL (**a**) and GroES, ATP-induced dissociation (**b**). The kinetics was derived from changes in fluorescence intensity of FAM-α-lactalbumin under conditions described in Legend to Figure 3. The dashed line represents an exponential approximation of the kinetics.

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
