# Peer review of "Back to GroEL-Assisted Protein Folding: GroES Binding-Induced Displacement of Denatured Proteins from GroEL to Bulk Solution"

_biomolecules, 2020, doi:10.3390/biom10010162_

Round 1
Reviewer 1 Report
The authors examined the location of denatured proteins priorly bound with GroEL-chaperone after GroES-co-chaperonin action in the presence of Mg-ATP using fluorescein-labeled denatured proteins. The results indicated that the GroES-induced transfer of the denatured proteins from GroEL to bulk solution occurs in a much shorter time than that proposed for the GroEL ATPase cycle.
This is an important study focusing on the mechanisms of protein folding which is required for exerting expected protein functions. As abnormalities of protein folding is a crucial step among the process of many neurodegenerative diseases, this manuscript will attract broad range of readers from basic researchers to physicians. The manuscript is well written.
Although I do not have any critical comments, minor suggestions to strengthen this manuscript are raised as follows:
Scientific interest of this manuscript is high. An issue regarding the significance of protein folding and misfolding in the mechanisms of diseases will increase clinical interest. For example, transthyretin amyloidosis is a representative protein misfolding disease because the structure of causative protein has been well clarified and novel disease modifying therapies now appear one after another (Biomedicines 2019; 7: E11). I would recommend incorporating this issue in the beginning of the introduction section, citing this article.
As an abbreviation “BRET” is spelled out in line 90, “bioluminescence resonance energy transfer (BRET)” in line 183 should be “BRET”.
“BRET” in figure 3 should be spelled out in figure 3 legends.
Author Response
Response to Reviewer 1 Comments
Dear Reviewer,
Thank you very much for your valuable comments aimed to improve our manuscript. We have addressed all of them, and our point-by-point responses are below; also, we have spell-checked the text.
Point 1: Scientific interest of this manuscript is high. An issue regarding the significance of protein folding and misfolding in the mechanisms of diseases will increase clinical interest. For example, transthyretin amyloidosis is a representative protein misfolding disease because the structure of causative protein has been well clarified and novel disease modifying therapies now appear one after another (Biomedicines 2019; 7: E11). I would recommend incorporating this issue in the beginning of the introduction section, citing this article.
Response 1: We have added the reference [Koike, H.; Katsuno, M. Biomedicines 2019, 7, 11] concerning the transthyretin amyloidosis and some discussion to Introduction (lines 40-46).
Point 2: As an abbreviation “BRET” is spelled out in line 90, “bioluminescence resonance energy transfer (BRET)” in line 183 should be “BRET”.
Response 2: Done.
Point 3: “BRET” in figure 3 should be spelled out in figure 3 legends.
Response 3: Done (see Figure 4).
Reviewer 2 Report
In the literature, two models of folding have been proposed, an active model that implies a direct involvement of GroEL and a passive model suggesting that folding occurs in the central cavity of the GroEL-GroES chaperonin complex.
The authors already proposed in another publication published in 2009, an alternative passive mechanism that does not require a cavity.
In this manuscript, the authors investigated this issue by studied the refolding of small (less than 30 KDa) denatured proteins such as pepsin, alpha-lactalbumin, serum albumin, lysozyme. Malate dehydrogenase. The main experimental strategy is based on the use of the so-called ‘streptavidin trap’. The conclusion of this study is that Mg-ATP and GroES induce the displacement of denatured proteins from GroEL to the bulk solution. The overall experimental set-up is appropriate and the experiments are conclusive. However, the manuscript requires a few minor specific improvements before full acceptance.
Major concerns:
- In the introduction, more recent references on the structure of GroEL structures are required.
- Fig1: can we visualize the displaced denatured fluorescent proteins?
- Table1: these results should be represented as a plot with curves to be more informative.
- Fig2: the figure legend is not clear, indicate precisely to what each curve correspond. What are the dashed lines?
- Fig3: The difference due to the addition of GroES compared to the panel without GroES is rather modest. Since no statistics are shown, this reviewer suggests performing replicas in order to add statistical analysis on the histogram. Moreover, a little cartoon depicting the assay would help the reader. According to the authors, the same results was obtained with malate-dehydrogenase, these complementary results should be included in this figure.
Author Response
Response to Reviewer 2 Comments
Dear Reviewer,
Thank you very much for your valuable comments aimed to improve our manuscript. We have addressed all of them, and our point-by-point responses are below; also, we have spell-checked the text.
Point 1: In the introduction, more recent references on the structure of GroEL structures are required.
Response 1: We have introduced the most recent review by U. Hartl et al. (2016) describing the GroEL structure and functions as a reference [17].
Point 2: Fig1: can we visualize the displaced denatured fluorescent proteins?
Response 2: Now it is clear from the text (lines 165-167) why we cannot show the displaced denatured fluorescent proteins in Fig. 1.
Point 3: Table1: these results should be represented as a plot with curves to be more informative.
Response 3: Done (see Fig. 2).
Point 4: Fig2: the figure legend is not clear, indicate precisely to what each curve correspond. What are the dashed lines?
Response 4: We have modified Legend to Fig. 3. Now it describes what dashed lines denote and to what measurement each curve corresponds (lettering from (a) to (d)).
Point 5: Fig3: The difference due to the addition of GroES compared to the panel without GroES is rather modest. Since no statistics are shown, this reviewer suggests performing replicas in order to add statistical analysis on the histogram. Moreover, a little cartoon depicting the assay would help the reader. According to the authors, the same results was obtained with malate-dehydrogenase, these complementary results should be included in this figure.
Response 5: In the revised version, Fig. 4 and its Legend have been modified to present the MDH data and some statistical analysis on the histograms. We also offer (lines 220-224) the possible reason for a rather modest difference between the panels with (complex SAVAq • fluorescent protein is present) and without GroES (the complex is absent). Besides, we have introduced a cartoon depicting the assay and events during the experiment with the “streptavidin trap” (see Figure 5).